



# Satellite Microwave Assessment of Northern Hemisphere Lake Ice Phenology from 2002 to 2015

Jinyang Du[1], John S. Kimball[1], Claude Duguay[2], Youngwook Kim[1],Jennifer D. Watts[1]

[1] Numerical Terradynamic Simulation Group, College of Forestry & Conservation, The University of Montana, Missoula, MT 59812, United States
[2] Department of Geography & Environmental Management and Interdisciplinary Centre on Climate Change, University of Waterloo, Waterloo, Ontario N2L 3G1, Canada

*Correspondence to*: Jinyang Du (jinyang.du@ntsg.umt.edu)

**Abstract.** A new automated method for satellite assessment of seasonal lake ice phenology at 5-km resolution was developed for all lake pixels (water coverage ≥ 90%) in the Northern Hemisphere using 36.5 GHz, H-polarized brightness temperature ($T_b$) observations from the Advanced Microwave Scanning Radiometer (AMSR-E/2) sensors. The lake phenology metrics include seasonal timing and duration of annual ice cover. A Moving *t*-Test (MTT) algorithm allows for automated lake ice retrievals with daily temporal fidelity and 5-km resolution gridding. The resulting ice phenology record shows strong agreement with available ground-based observations from the Global Lake and River Ice Phenology Database (95.4% temporal agreement), and favourable correlations ($R$) with alternative ice phenology records from the Interactive Multisensor Snow and Ice Mapping System ($R = 0.84$ for water clear of ice [WCI] dates; $R = 0.41$ for complete freeze over [CFO] dates) and Canadian Ice Service ($R = 0.86$ for WCI dates; $R = 0.69$ for CFO dates). Analysis of the resulting 12-year (2002-2015) AMSR ice record indicates increasingly shorter ice-cover duration for 43 out of 71 (60.6%) Northern Hemisphere lakes examined, with significant (p < 0.05) regional trends toward earlier ice melting for only five lakes. Higher latitude lakes reveal more widespread and larger trends toward shorter ice cover duration than lower latitude lakes, consistent with enhanced polar warming. This study documents a new satellite-based approach for rapid assessment and regional monitoring of seasonal ice cover changes over large lakes, with resulting accuracy suitable for global change studies.

## 1 Introduction

Ice phenology describes the seasonal cycle of lake ice cover and encompasses freeze-up and break-up periods, and ice cover duration (Duguay et al., 2015a). Freeze-up corresponds to the time period between the beginning of ice formation and the formation of a complete sheet of ice; break-up involves the time period between the onset of spring melt and the complete disappearance of ice from the lake surface (Kang et al., 2012). These ice phenology variables are key metrics sensitive to weather and climate conditions, and influencing lake-atmosphere interactions and hydrological and ecological processes in high-latitude and high-altitude regions (Duguay et al., 2006; Mishra et al., 2011; Duguay et al., 2012; Duguay et al., 2015a). By insulating lake water from the overlying atmosphere and minimizing water and atmosphere heat and gas exchanges, lake



ice has a controlling influence on water-column oxygen concentration, water temperature and the composition and abundance of aquatic species (Livingstone, 1997; Bengtsson and Herschy, 2012; Kang et al., 2012; Wrona et al., 2016). In addition to the impacts on aquatic life, the formation and disappearance of lake ice also has a significant influence on the spread of man-made pollutants such as perfluorinated chemicals (PFCs) (Veillette et al., 2012; Wrona et al., 2016). The extent and duration of lake ice also affect human activities, including hydroelectric power generation, navigation and winter transportation, and production and distribution of food and water (Schröter et al., 2005; Weyhenmeyer et al., 2011). Moreover, lake ice phenology is closely coupled with atmospheric heat fluxes (Latifovic and Pouliot, 2007; Park et al., 2016) and sensitive to the alteration of weather patterns under projected global warming (Magnuson et al., 2000).

Accurate and consistent records of lake ice phenology especially in data-sparse regions, including much of the pan-Arctic and Qinghai-Tibetan Plateau regions, provide valuable information for monitoring global change impacts on high-latitude and high-altitude environments (Magnuson et al., 2000). Previous studies have documented significantly earlier ice break-up between 1951 and 2000 for lakes in Canada (Duguay et al., 2006) and decreasing lake ice cover of Lake Ladoga in Europe during the last few decades (Karetnikov and Naumenko, 2008). Shorter ice-cover seasons may promote greater $CH_4$ emissions from northern lakes (Greene et al., 2014) which could reinforce further climate warming due to the role of $CH_4$ as a potent greenhouse gas. Despite a general tendency for later freezing and earlier break-up in the Northern Hemisphere (Magnuson et al., 2000), various tendencies over specific lakes and time periods may exist. Observations from satellite altimetry and radiometry over 1992–2004 for Lake Baikal showed a tendency for colder winters, with earlier ice formation, later ice break-up, and ice duration increase (Kouraev et al., 2007a, 2007b).

A historic lake ice phenology database has been assembled from long-term ground-based observations across the northern domain (Magnuson et al., 2000; Benson et al., 2012); however, the number of monitoring sites is extremely sparse, with variable observational recording periods and methods, which limits capabilities for regional assessment and monitoring of environmental changes. Data acquired from space-borne optical-thermal infrared (TIR) and microwave sensors have also been applied for monitoring river and lake ice phenology at regional and global scales (Chu et al., 2016). Optical-TIR remote sensing can provide accurate estimation of land surface temperature (LST) and classification of land cover types at relatively fine spatial resolutions (~10 s to 100 s of meters), while LST (1 km resolution) and snow cover (500 m resolution) products derived from MODIS (Moderate Resolution Imaging Spectroradiometer) have been used to infer lake ice conditions (Nonaka et al., 2007; Hall et al., 2010; Kheyrollah Pour et al., 2012). Time series of AVHRR (Advanced Very High Resolution Radiometer) imagery have also been used to classify Canadian lake ice phenology events with relatively high accuracy (Latifovic and Pouliot, 2007). However, regional monitoring of lake ice dynamics from satellite optical-TIR sensors is constrained by signal degradation and data loss stemming from seasonal reductions in solar illumination at higher latitudes and persistent cloud cover, smoke and other atmospheric aerosol contamination.

Satellite microwave remote sensing at lower frequencies (~ < 89 GHz) is relatively insensitive to solar illumination and atmosphere constraints, while current microwave radiometers on-board polar orbiting satellites provide frequent (~ daily) observations spanning northern (≥45°N) land areas. The active and passive microwave retrievals are also highly sensitive to



the large contrast in surface dielectric properties between open water and ice cover over large lakes. Despite successful applications using active microwave remote sensing in lake ice retrieval (Leconte and Klassen, 1991; Nolan et al., 2002; Howell et al., 2009; Geldsetzer et al., 2010), capabilities for global lake ice monitoring from satellite radar sensors have been constrained by limited global coverage and temporal frequency of observations. Alternatively, space-borne microwave radiometers provide long-term (since 1978) brightness temperature ($T_b$) observations with relatively high temporal fidelity (~1-2 days) especially at higher ($\geq 45°N$) latitudes. The satellite $T_b$ retrievals are capable of detecting lake ice phenology events coinciding with large changes in surface emissivity, but the passive microwave retrievals are constrained by a generally coarser spatial resolution than radar and optical-TIR sensors. Despite these limitations, ice freeze-up and break-up events for Great Slave Lake (GSL) were monitored using a threshold-based method for SSM/I (Special Sensor Microwave Imager) observations at 85 GHz (Walker and Davey, 1993; Ménard et al., 2002). Recently, H-Polarized AMSR-E (Advanced Microwave Scanning Radiometer for EOS) $T_b$ observations at 19 GHz were analyzed to determine ice phenology for GSL and Great Bear Lake (GBL), the two largest lakes in northern Canada (Kang et al., 2012). Similar $T_b$ records from SSM/I and SMMR (Scanning Multichannel Microwave Radiometer) were also used to monitor lake ice phenology for Nam Co Lake (Ke et al., 2013) and Qinghai Lake (Che et al., 2009) within the high elevation Qinghai-Tibetan Plateau. Previous studies based on satellite passive microwave remote sensing have mainly focused on one or two lakes using empirical algorithm thresholds developed for specific study areas. There is a great need to develop a universal lake ice detection algorithm and establish a consistent ice phenology database covering major lakes at the global scale for climate impact assessments such as those published by the Intergovernmental Panel and Climate Change (Stocker et al., 2013).

In this study, we present a new automated method to derive lake ice phenology using 36.5 GHz H-polarized satellite radiometric $T_b$ measurements from AMSR-E and AMSR2 (Advanced Microwave Scanning Radiometer 2). The algorithm is used to produce daily lake ice maps with 5 km gridding. The resulting AMSR Lake Ice Phenology (LIP) record encompasses all 5 km by 5 km lake pixels (water coverage $\geq$ 90%) within the Northern Hemisphere ($\geq 0$ °N) and spans more than 12 years of observations encompassing both AMSR-E (June 2002 to September 2011) and AMSR2 (June 2012 to December 2015) satellite sensor records. Here we present a detailed methods description and evaluation of the LIP product against other independent observations and alternative lake ice products. A trend analysis is also conducted to characterize recent regional LIP changes over the study period.

## 2 Methods

### 2.1 Study domain and datasets

#### 2.1.1 Study domain

Accurate monitoring of seasonal land freeze/thaw and lake freeze-up/break-up events which are widespread in the Northern Hemisphere has been recognized as an essential component for understanding interactions and feedbacks between regional



ecosystems and climate change (Duguay et al., 2006; Kim et al., 2012; Du et al., 2015a). This study utilizes satellite passive microwave remote sensing to detect lake ice changes for 5-km lake pixels in the Northern Hemisphere, with a particular focus on lake ice phenology in the mid- and high- latitudes (≥ 30° N). The domain (Fig.1) includes the high northern pan-Arctic region and high altitude Qinghai-Tibetan Plateau, which are data-sparse but strongly sensitive to global warming (Woo et al., 2007; Wang et al., 2011). Both regions are also characterized by cold climate conditions with extensive winter ice cover. The resulting domain includes three sets of lakes for algorithm evaluation and lake ice phenology analysis. Among the lakes analyzed, four are represented in the Global Lake and River Ice Phenology Database (GLRIPD) (Benson and Magnuson, 2000) and were used to evaluate the LIP estimates on a per pixel basis against available ground-based observations; the four GLRIPD lakes evaluated include Lake Superior in the USA; Lake Oulujarvi, Lake Haukivesi and Lake Paijanne in Finland. In addition, 12 North American lakes (GBL, GSL, Smallwood Lake, Nettiling Lake, Dubawnt Lake, Amadjuak Lake, Wollaston Lake, Baker Lake, Kasba Lake, Lesser Slave Lake, Red Lake, Peter Pond Lake) that experience annual break-up and freeze-up events were selected for lake-wide inter-comparisons between the LIP metrics derived from this study and alternative lake ice products from the Interactive Multisensor Snow and Ice Mapping System (IMS) (Helfrich et al., 2007; http://www.natice.noaa.gov/ims/) and the Canadian Ice Service (CIS) (Howell et al., 2009). Finally, regional LIP trends were assessed over the 12 year (2002-2015) satellite record for 71 Northern Hemisphere lakes identified in the Global Lakes and Wetlands Database (GLWD) (Lehner and Döll, 2004). The lakes selected encompass approximately 23% (297,044 km$^2$) of the estimated total surface area of large lakes (area ≥ 50 km$^2$) within the domain (Lehner and Döll, 2004). The 71 lakes were selected on the basis of having: (a) at least one 5-km lake pixel with 100% water coverage and located outside of a 5-km land buffer zone; and (b) all pixels representing the lake having at least 20 days with full ice coverage and 20 days with open water. Criterion (a) was used to reduce potential contamination from adjacent land areas since the native resolutions of 36.5 GHz observations are approximately 14 km×8 km for AMSR-E and 12 km×7 km for AMSR2, respectively (Imaoka et al., 2010). Criterion (b) emphasizes lakes having extended ice and open water seasons rather than those with a temporary ice cover or short open water phase. The 20-day minimum duration was set according to the pre-defined subsample sizes of our algorithm (Section 2.3).

**2.1.2 Datasets used for algorithm development**

The lake ice detection algorithm developed in this study relies primarily on 36.5 GHz H-polarized $T_b$ retrievals from AMSR-E and AMSR2. The AMSR-E sensor was operational on the NASA Aqua satellite from June 2002 to October 2011 and provided twice-daily measurements of global microwave emissions over land with descending/ascending orbital equatorial crossings at 1:30 AM/PM local time, and vertically (V) and horizontally (H) polarized $T_b$ retrievals at six frequencies (6.9, 10.7, 18.7, 23.8, 36.5, 89.0 GHz). For this study, we used the AMSR-E ascending 36.5 GHz orbital swath data at the native footprint resolution of approximately 14 km×8 km (Kawanishi et al., 2003). After the cessation of AMSR-E operations on 4 October 2011, its successor AMSR2 was launched on 18 May 2012 on-board the sun-synchronous JAXA GCOM-W1 satellite. AMSR2 is similar to AMSR-E in sensor configuration, including frequencies, incidence angle and orbital equatorial





crossing time. Major AMSR2 advancements over AMSR-E include an additional frequency at 7.3 GHz designed for mitigating Radio Frequency Interference (RFI) and a larger (2.0 m diameter) main reflector for enhanced spatial resolution. The AMSR2 L1R (version 1.2) re-sampled ascending swath 36.5 GHz $T_b$ retrievals at approximately 12 km×7 km resolution were used for this study. The un-calibrated AMSR2 $T_b$ retrievals were estimated to be positively biased against AMSR-E by

~ 1.3 K (Du et al., 2014). However, the sensor inconsistency is expected to have minimal impacts on our algorithm, which relies on $T_b$ time series change signal detection rather than $T_b$ absolute accuracy.

The Level 1 Global Lakes and Wetlands Database (GLWD) (Lehner and Döll, 2004) comprises the 3,067 largest lakes (area $\geq$ 50 km$^2$) and 654 largest reservoirs (storage capacity $\geq$ 0.5 km$^3$) worldwide and was used for identifying Northern Hemisphere water bodies and the 71 large lakes used for the LIP assessment (Fig. 1). We also used the MODIS 250 m land-

water mask (MOD44W) for calculating the proportional water coverage of 5-km resolution pixels within lake areas identified by the GLWD (Carroll et al., 2009).

### 2.1.3 Datasets used for algorithm evaluation

Four lake ice phenology databases were used to evaluate the LIP retrievals including (a) the GLRIPD (Benson and Magnuson, 2000); (b) the National Oceanic and Atmospheric Administration (NOAA) IMS 4-km daily snow and ice product

(Helfrich et al., 2007; http://www.natice.noaa.gov/ims/); (c) the CIS lake-wide ice product (Howell et al., 2009); and (d) MODIS quick-look images for GBL downloaded from the Geographic Information Network of Alaska (http://www.gina.alaska.edu).

The GLRIPD contains descriptive ice cover data for 865 lakes and rivers in the Northern Hemisphere (Benson and Magnuson, 2000). The GLRIPD includes ground-based (lakeshore) observations that were used for evaluating the

corresponding LIP results for the targeted lakes. The GLRIPD records the first date when the water body was observed to be completely ice covered, and the date when the last ice break-up was observed before the summer open water phase for each year of record (Benson and Magnuson, 2000). For evaluating LIP results representing 5-km lake dominant pixels (water coverage $\geq$ 90%), the lake was assumed completely covered with ice for the period between the first date with complete ice cover and last ice break-up date as recorded in the GLRIPD, while lakes were classified as open water condition for other

dates within each annual cycle. Only four lakes, including Lake Superior, USA, Lake Oulujarvi, Lake Haukivesi, and Lake Paijanne in Finland, were selected for the LIP comparisons due to a predominance of ice observations from smaller lakes in the GLRIPD database. The temporal coverage of GLRIPD observations for the four lakes that overlaps with the AMSR record extends from 2002 to 2007 for Lake Superior and 2003 to 2007 for Lake Oulujarvi, Lake Haukivesi, and Lake Paijanne, respectively.

The NOAA IMS daily snow and ice product provides snow and ice cover extent information derived from ground observations and an extensive variety of satellite observations, including AVHRR, GOES (Geostationary Operational Environmental Satellite), SSM/I and AMSU (Advanced Microwave Sounding Unit) (Helfrich et al., 2007). The CIS lake-ice product estimates lake ice cover fraction in tenths (0: open water – 10: complete ice cover) for nearly 140 lakes across



Canada and the northern USA from visual interpretation of 1.1 km resolution NOAA AVHRR and 100 m resolution RADARSAT ScanSAR imagery (Howell et al., 2009), and MODIS (250-500 m) and Visible/Infrared Imaging Radiometer Suite (VIIRS) I-Band (375 m) observations. The CIS product provides a single lake-wide value per lake on a weekly basis. Both the 4-km IMS grid products (year 2004-2015) and CIS data (year 2002-2015) were used for lake-wide comparisons

against the resulting LIP retrievals.

MODIS quick-look images in true-color composites (Bands 1, 4, 3 in RGB) were selected for qualitative visual comparisons with the LIP results. The MODIS images were acquired over the break-up season (2012-2013) with clear-sky conditions on June 22, June 27, and July 8, 2013 and extensive cloud cover on July 5, 2013. The quick-look products were provided at 250-m resolution in Albers Equal-area Conic projection.

In addition, ERA-Interim (Dee et al., 2011) quarter-degree reanalysis surface air temperature (SAT) data was analyzed for evaluating LIP trends over the 71 Northern Hemisphere lakes selected (Fig.1). ERA-Interim is a global atmospheric model data reanalysis produced by the European Centre for Medium-Range Weather Forecasts, and the data assimilation system used to produce ERA-Interim is based on a 2006 release of the IFS (Cy31r2) including a 4-dimensional variational analysis (4D-Var) with a 12-hour analysis window (Dee et al., 2011).  Daily average SAT over the spring (MAM) and fall (SON)

seasons of years 2002 to 2015 was extracted for the quarter-degree grids encompassing the lake centers.

### 2.1.4 Data processing

To derive the LIP estimates, AMSR-E/2 36.5 GHz orbital swath $T_b$ data were spatially re-sampled to a 5 km resolution polar EASE-Grid (version 2) format using an inverse distance squared weighting method (Ashcroft and Wentz, 2000; Brodzik et al., 2012; Brodzik et al., 2014). It is worth noting that the $T_b$ spatial gridding is posted at 5 km resolution while the original

36.5 GHz AMSR observations have coarser native sensor footprints (~12 km for AMSR-E and 9 km for AMSR2). Before carrying out the $T_b$ gridding process, an additional altitude correction was made to the AMSR-E data by considering the actual surface of the Earth instead of that of an ideal Earth ellipsoid. The same altitude correction was used for the AMSR2 L1R data (Maeda et al., 2016). According to Maeda et al. (2016), an altitude of 3,000 m leads to about 4 km displacement of AMSR2 $T_b$ geo-location. Thus the altitude correction is a necessary prerequisite to ensure reliable analysis of AMSR-E/2

lake ice phenology retrievals at higher elevations, including the Qinghai-Tibetan Plateau.

The finer resolution MOD44W static open water maps were aggregated to the same 5 km resolution polar EASE-grid 2.0 projection format as LIP and used with the GLWD to identify dominant lake pixels (water coverage ≥ 90%) where the AMSR lake ice detection was made. The 250-m resolution MODIS quick-look images were re-projected to the EASE-grid 2.0 projection for visual comparisons with the LIP results.

### 2.2 Algorithm theoretical basis

Accurate modeling of satellite observed microwave emissions from lakes is complex and requires good understanding of microwave scattering and emitting mechanisms from atmosphere and lake elements. Microwave emissions from a non-



scattering atmosphere are governed by both air temperature and atmosphere optical thickness, which is approximately the sum of the optical thickness of oxygen, cloud liquid water, and atmospheric water vapor (Wang and Tedesco, 2007; Du et al., 2015b). Microwave emissions from a lake with an upper layer that may consist of water, ice and snow, are determined by a number of factors; these factors include lake surface roughness, water dielectric properties mainly affected by water salinity

and temperature, ice thickness and dielectric properties, snow cover dielectric properties mainly controlled by snow density and wetness, snow particle size and stratification of snow and ice layers (Du et al., 2010; Lemmetyinen et al., 2010; Lemmetyinen et al., 2011). Despite the complexity of the lake emission problem, sharp changes in satellite microwave $T_b$ observations at multiple frequencies are evident during the transitions between lake freeze-up and break-up periods. For example, previous studies showed low $T_b$ measurements (< 150 K) from H-polarized 37-GHz SMMR data over low

emissivity open water regions of the Great Lakes and Gulf of Mexico, contrasting with much higher $T_b$ values (> 215 K) over western Lake Superior under frozen conditions due to the high emissivity of lake ice (Ferraro et al., 1986). Similarly, the H-polarized emissivity at 35 GHz and 50 degree incidence angle is approximately 0.356 for a calm and unfrozen lake at 0-8 degrees Celsius, and is well below the emissivity (> 0.610) of different types of snow and ice (Mätzler, 1994). These studies suggest a very large Ka-band $T_b$ difference (> 60 K) between a lake at 0 ℃ with no ice and 100% ice coverage. The

timing of ice formation and disappearance can therefore be determined by the large characteristic $T_b$ changes indicated from satellite passive microwave observations (Walker and Davey, 1993; Che et al., 2009; Kang et al., 2012).

**2.3 Algorithm development**

For identifying freeze-up and break-up events, a Moving $t$-Test method (MTT) was introduced to detect abrupt temporal changes in the H-Polarized 36.5 GHz $T_b$ observations from AMSR-E and AMSR2. Selection of 36.5 GHz $T_b$ observations

from other available AMSR frequencies represents a compromise between finer spatial resolutions gained from higher frequencies and less sensitivity to potential atmosphere contamination available from lower $T_b$ frequency observations. Moreover, H-Polarization $T_b$ retrievals were used instead of V-Polarization data due to their reported higher sensitivity to lake freeze-up/break-up signals (Kang et al., 2012). The detailed lake ice detection method used in this study is described below.

*Step1: Detection of abrupt changing point*

The MTT method was initially developed for detecting abrupt climate changes by examining whether the difference between the mean values of two subsamples is statistically significant (Jiang and You, 1996; Xiao and Li, 2007). As detailed in the literature (Xiao and Li, 2007), for a time series with $n$ elements, a $t$-test is made at each point $x_k$ for evaluating the difference of the two subsets $x_{k1}$ and $x_{k2}$ ($n_1 \leq k \leq n - n_2$, and $n_1$, $n_2$ are the sub-sample sizes) before and after $x_k$. The $t$-

statistic is defined as:

$$t = \frac{\overline{x_{k2}} - \overline{x_{k1}}}{s_k \sqrt{\frac{1}{n_1} + \frac{1}{n_2}}} \qquad (1)$$





where $s_k = \sqrt{\frac{n_1 s_{k1}^2 + n_2 s_{k2}^2}{n_1 + n_2 - 2}}$ , $\overline{x_{k2}}$ and $\overline{x_{k1}}$ are the mean values, and $s_{k1}^2$ and $s_{k2}^2$ are the variances for the two subsets, respectively. Given a significance level $\alpha$, $x_k$ is determined as an abrupt changing point if $|t| \geq t_\alpha$. In this study, we define $\alpha$ as 0.005 and temporal sub-sample sizes as $n_1 = n_2 = 20$ days. The 20-day requirement is set for excluding potentially dynamic $T_b$ changes caused by short-term weather events such as storms.

*Step2: Determining reference $T_b$ values for lake ice conditions*

For a group of detected changing points sequenced from $p$ to $q$, the mean $T_b$ values $\overline{x_{p1}}$ and $\overline{x_{q2}}$ as defined in Eq. (1) are representative of the satellite observations over the stable stages before and after the changing period, respectively. For a lake experiencing a complete annual freeze-up/break-up cycle, at least two groups of seasonal changing points can be defined. Besides the sharp $T_b$ increases as lake water freezes, the melting of dry snow overlying lake ice to wet snow can

induce further increases in the observed $T_b$ since microwave emissions from wet snow are close to that of a blackbody (Ulaby et al., 1986). Therefore, assuming $\overline{x_{p1}}$ is always smaller than $\overline{x_{q2}}$ , we define the lowest $\overline{x_{p1}}$ of all changing groups as the reference $T_b$ for lake water and $\overline{x_{q2}}$ from the same group as the reference $T_b$ for lake ice. Lake ice conditions for a given date $i$ can thus be determined as:

Ice dominant if $T_{b_i} \geq T_{b_{threshold}}$

Water dominant if $T_{b_i} < T_{b_{threshold}}$  (2)

where $T_{b_i}$ is the $T_b$ for date $i$, $T_{b_{threshold}} = (\overline{x_{p1}} + \overline{x_{q2}})/2.0$ and $\left|\overline{x_{p1}} - \overline{x_{q2}}\right|$ is required to be larger than 30 K since liquid/ice phase changes of lake water can lead to large $T_b$ changes exceeding 60 K as introduced in Section 2.2.

*Step3: Deriving lake ice status*

Based on Eq. (2), lake ice status is first derived for each point $i$ in the $T_b$ times series where $n_1 \leq i \leq n - n_2$ using a

temporally smoothed $T_{b_i}$ defined as the mean $T_b$ within the range $[i - n_1/2, i + n_2/2]$. The use of a smoothed $T_{b_i}$ minimizes the impact of high temporal frequency events in the time series while emphasizing lower frequency lake ice-covered and ice-free signals. Thus for point $j$ whose temporally adjacent points have different lake ice status, the refined lake ice detection is carried out using Eq. (2) for each observed $T_b$ value within the range $[j - n_1/2, j + n_2/2]$. The above lake ice detection process was carried out for each $T_b$ time series from AMSR-E and AMSR2 separately because of the 7-month gap (Oct 4,

2011 – May 18, 2012) in the observation records between the two sensors. For running the algorithm, missing daily $T_b$ retrievals were gap-filled through temporal linear interpolation of adjacent successful $T_b$ retrievals acquired from the same ascending orbits. However, only the lake ice detection results corresponding to the actual satellite observations were output for further analysis.

The above MTT algorithm was applied to all 5-km pixels with dominant ($\geq 90\%$) open water coverage within the Northern

Hemisphere domain on a daily basis to generate the AMSR LIP dataset describing lake ice conditions. The dominant ($\geq 90\%$) open water coverage criterion is set to include lake pixels while reducing potential contamination from adjacent land areas.



### 2.4 Evaluation of lake ice phenology retrievals

The resulting LIP retrievals were evaluated against other available lake ice databases, including the GLRIPD, IMS and CIS products. The remotely sensed lake ice phenology variable definitions from this study are summarized in Table 1 relative to the other lake ice observational data records used for LIP validation on a per pixel basis and for entire lakes (Kang et al., 2012; Duguay et al., 2015a).

For comparing with the GLRIPD ground-based records, the LIP ice-on and ice-off dates were extracted for the 5-km pixel closest to the GLRIPD observation site. The pixel representing Lake Superior has 100% water coverage (Lat/Lon: 46.78°N/-90.45°W). For the pixels representing Lake Oulujarvi (Lat/Lon: 64.3°N/27.3°E), Lake Haukivesi (Lat/Lon: 62.07°N/28.57°E), and Lake Paijanne (Lat/Lon: 61.19°N/25.55°E), the water coverage is 100%, 91.4% and 95.7%, respectively.

The LIP derived annual CFO and WCI dates for the 12 selected lakes were also compared with alternative IMS and CIS ice products. Different from the dominant open water coverage ($\geq 90\%$) requirement set for generating the LIP database, only lake pixels with complete (100%) open water coverage and outside of the 5-km land buffer zone were considered in the IMS and LIP comparisons; this same criterion was set for the lake-wide comparisons to minimize potential contamination from adjacent land areas since the native 36.5 GHz AMSR footprint ranges from approximately 9-12 km. Considering possible retrieval uncertainties, the CFO/WCI dates derived from the LIP and IMS datasets were slightly adjusted from the definitions in Table 1 and were determined as the dates when most (99.5% for this study) lake pixels were identified as ice/water. The CIS CFO dates were determined when the reported lake ice fraction was 9 followed by changes from 9 to 10; and WCI dates were derived when the reported lake ice fraction was 1 followed by changes from 1 to 0. The derived CIS CFO/WCI dates are comparable with corresponding LIP results, excluding waters adjacent to land such as part of eastern arm of the GSL where a high concentration of islands exists and ice formation/melting timing was found to be different from other GSL areas (Howell et al., 2009; Kang et al., 2012).

### 2.5 Analysis of lake ice phenology changes

Based on the LIP database covering the AMSR-E (Jun., 2002 - Oct., 2011) and AMSR2 (Jun., 2012 – Dec., 2015) observation periods, we selected 71 lakes in the Northern Hemisphere from 250 of the world's largest lakes (including both natural and artificial lakes), as described in the GLWD, to analyze potential lake ice phenology trends, including WCI date, CFO date and annual ICDe. In order to assess the pattern of recent Northern Hemisphere lake ice phenology changes, a temporal trend analysis was performed on the 12-year LIP record for each of the 71 lakes. The assumption of independent observations was first determined using a correlogram (Noguchi et al., 2011). For ice phenology time series without significant autocorrelation detected, the magnitude and significance of temporal trends were tested using the non-parametric Mann-Kendall and Sen's methods (Sen, 1968; Duguay et al., 2006). Alternatively, for a time series with persistent serial correlation, additional pre-whitening approaches were applied (Zhang et al., 2000). For evaluating LIP derived lake



phenology, a similar temporal trend analysis was also carried out on the ERA-Interim daily average SAT over the lakes for the spring and fall seasons from 2002 to 2015.

## 3 Results

### 3.1 LIP comparisons with GLRIPD lake observations

The lake ice status derived from the LIP and GLRIPD records are plotted for the selected large lake validation sites (Fig. 2), including Lake Superior (a), Lake Oulujarvi (b), Lake Haukivesi (c), and Lake Paijanne (d), along with the daily ascending AMSR $T_b$ retrievals. The LIP results show generally strong agreement with the GLRIPD site observations of lake ice conditions for the four lakes examined, with overall retrieval accuracy of 95.4%. The lake ice/water retrieval error at the beginning of the record for Lake Haukivesi (Fig.2c) may be caused by partial melting of lake ice in January and February
2003 that resulted in low AMSR $T_b$ observations. While the AMSR $T_b$ observations show dynamic daily fluctuations due to changing water and atmosphere properties (Section 2.2), lake freeze-up and break-up events constitute the dominant factors affecting seasonal $T_b$ changes. The effects of higher temporal frequency $T_b$ variations are minimized in the LIP algorithm by the pre-defined 20-day subsample sizes (Section 2.3), which represent a compromise between the algorithm's capability in capturing shorter-term lake ice formation or melt events, and potentially degraded lake ice/water seasonal retrieval accuracy.

### 3.2 LIP comparisons against MODIS imagery, IMS and CIS products

An example visual comparison between the LIP results and MODIS quick-look imagery (Fig. 3) shows the 2013 spring ice break-up process over the GBL. In this example, both datasets show a general onset of lake ice breakup on Jun. 22 (Fig.3 a), similar spatial ice distribution patterns on Jun. 27 (Fig.3 b) and ice-free conditions on Jul. 8[th] (Fig.3 d). Despite extensive cloud presence in the MODIS image for Jul. 5[th], both MODIS and LIP indicate remaining ice cover on the western edge of
GBL (Fig.3 c). A few remaining pixels along the GBL coast line were identified as ice-covered in the LIP results for Jul. 8[th] (Fig.3 d); the apparent LIP retrieval error is attributed to land contamination, while the affected pixels are within the 5 km land buffer zone (Section 2.4) and excluded in the final LIP product.

The LIP products were compared against similar lake ice phenology metrics from the IMS and CIS datasets for the 12 North American study lakes. For GBL and GSL, the LIP products agreed well with the CIS records; temporal correlations
($R$-value) of 0.90 and 0.85 were observed for the WCI dates for GBL (Fig. 4a) and GSL (Fig. 4b), respectively, while correlations of 0.90 and 0.72 were determined for GBL (Fig. 4c) and GSL (Fig. 4d) CFO dates. The LIP results were also strongly correlated with the IMS record on derived WCI dates for both lakes (R = 0.94 for GBL and R = 0.91 for GSL) (Figs. 4a, 4b). However, lower correspondence was found between LIP and IMS CFO dates, with respective correlations of 0.54 and 0.63 for GBL and GSL (Figs. 4c, 4d). These results indicate that the LIP derived lake ice phenology variables are
generally consistent with the IMS and CIS records for the two lakes examined, with generally higher (lower) correspondence for WCI (CFO) dates. For GBL, the LIP estimated CFO dates occur similar to the CIS records (0-day difference) and later



than the IMS records by about 3 days; the LIP WCI dates occur earlier than the CIS and IMS records by about 2 and 3 days, respectively. For GSL, the LIP record also shows earlier (later) CFO than the CIS (IMS) records by about 2 (3) days, and earlier WCI dates than both CIS and IMS by about 6 and 5 days, respectively. The inter-comparisons between CIS and IMS show average 5-day and 0-day differences in respective CFO and WCI dates for their overlapping period from 2004 to 2015

for GBL; corresponding differences for GSL are 5 days and 1 day, respectively. The differences between the LIP and CIS, and IMS metrics are of similar magnitude as the differences between the CIS and IMS metrics.

The LIP comparison results for all 12 study lakes are summarized in Table 2. Similar to the comparisons for GBL and GSL, the LIP results are strongly correlated with both CIS and IMS records for WCI dates, with respective average temporal correlations ($R$-value) of 0.86 and 0.84. For CFO dates, the average correlation between LIP and CIS results are also strong

($R = 0.69$) while only moderate correlation ($R = 0.41$) was found between the LIP and IMS results. The LIP estimated CFO dates tend to occur earlier than the CIS record by about 6 days and later than the IMS record by about 1 day. The LIP record also shows earlier WCI dates than the CIS and IMS records by about 7 and 2 days, respectively.

**3.3 Analysis of LIP lake ice phenology changes**

The magnitude and direction of LIP trends were calculated for the 71 Northern Hemisphere study lakes for the 12-year

AMSR record. Among all 71 lakes, 43 (60.6%) show declining trends in ICDe indicating an increasingly shorter ice-cover season, while the other lakes show either increasing or minimal change in annual ice cover (Fig. 5). However, no observed ice trends are statistically significant ($p \geq 0.05$). The lack of significant trends is attributed to large yearly variability (± 13.3 day) in average ICDe and a relatively short (12-year) LIP observation record. The changing trends also demonstrate a latitudinal pattern, as 81.0% of the lakes (17 out of 21) at higher latitudes (> 60°N) show declining ICDe trends while only

45.0% (9 out of 20) of lower latitude (< 50°N) lakes show a similar trend.

The observed changes in ICDe are the net result of changes in fall CFO and spring WCI dates. A tendency toward earlier WCI dates was found for 40 lakes, including 5 lakes (Lake Vygozero, Lake Barun-Torey, Lake Segozero, Novosibirsk Reservoir in Russia, and Lake Teshkpuk in USA) with significant LIP trends. However, no lakes showed significant trends toward later spring break-up. Similar to the ICDe analysis, most high-latitude lakes (81.0 % of lakes above 60°N) show

earlier spring thaw trends, while only 45.0 % of lower latitude (<50°N) lakes show similar trends. A tendency toward delayed CFO was found for 35 of the 71 lakes (49.3%) examined, but no trends are statistically significant ($p \geq 0.05$). Lake Bosten in China was the only lake with a significant trend toward earlier freeze-up. There was no clear relationship between changes in lake CFO dates with latitude.

Similar analysis of quarter-degree ERA-Interim SAT over the study lakes indicates a much stronger warming trend in

spring (0.073 °C yr$^{-1}$) than autumn (0.023 °C yr$^{-1}$). Moreover, similar to the latitudinal pattern shown in the LIP-based analysis, the SAT increase in the spring is positively correlated with latitude ($R = 0.33$; p = 0.005), while no SAT correlation with latitude is found for the autumn ($R \sim 0.0$).



## 4 Discussion

We developed a new satellite approach for regionally consistent classification of ice phenology for large lakes in the Northern Hemisphere from the AMSR sensor record. We used similar 36.5 GHz, H-polarization daily brightness temperature retrievals from AMSR-E and AMSR2 sensor records with 5 km posted spatial resolution. The resulting LIP record

documents the timing and duration of seasonal lake phenology events of 5-km lake pixels (water coverage ≥ 90 %) over the 12-year AMSR record. The LIP results showed strong agreement with GLRIPD site observations from four lakes, with agreement ranging from 92.4% to 98.7%. Differences between the LIP and GLRIPD results can be attributed to several factors. First, each database has a different definition of lake ice conditions; lake ice coverage determined by satellite microwave sensors is dependent on ice thickness, which may vary from the ice detection approach used by on-site observers

or observed from optical sensors. According to the literature (Hall et al., 1981), lake ice thickness and $T_b$ are linearly related for multiple frequencies (from 5 to 37 GHz). The reported maximum microwave penetration depths of fresh lake ice at 37 GHz ranges from 0.70 m to 1.4 m depending on ice temperatures (Chang et al., 1997; Surdyk, 2002; Kang et a., 2014). This implies that the formation of thin ice, resulting in relatively small $T_b$ increases, may not be detectable using the defined $T_b$ thresholds in the LIP algorithm. Differences between the LIP and GLRIPD results may also reflect spatial inconsistencies in

lake observation area between the ground-based lakeshore observations and the coarser satellite footprint. Thus, the lake area observed on-site may not completely overlap with the AMSR lake pixel used for the LIP classification. The GLRIPD also does not provide explicit descriptions of lake ice status for the period between the first date when the water body was completely ice covered and the date when the last ice break-up occurred; thus, short-term events such as temporary ice melting or formation may not be recorded in the GLRIPD. For example, though identified as ice-covered in the GLRIPD,

Lake Oulujarvi was more likely to have thawed on January 9, 2007 since the low $T_b$ (178.7 K) observation is more characteristic of open water emissions (Fig.2b). The satellite $T_b$ observations at 36.5 GHz are also affected by other factors than surface freeze-up/break-up transitions, including changes in atmosphere water vapor and cloud liquid water (e.g. Section 2.2). For example, the LIP incorrectly detected ice-on conditions for Lake Haukivesi on July 30, 2004 (Fig.2 c) likely due to prolonged high water vapor concentration in the midsummer resulting in a large $T_b$ increase similar to a

seasonal freeze-up event.

In the lake-based comparisons for the 12 lakes examined, including GSL and GBL, the LIP results show strong correspondence with the CIS product for both CFO and WCI dates, and similar high correlations with the IMS results for WCI dates; however, the LIP WCI (CFO) dates differ by approximately 7 (6) days from the CIS and 2 (1) days from the IMS. These differences are attributed to the different sensors, spatial/temporal resolutions and retrieval methods associated with

the different products. As described in Section 2.1.3, the CIS product is derived for individual lakes from visual interpretation of imagery from optical and SAR sensors, and has a ±1 week accuracy due to the weekly product reporting. Both CIS and IMS products rely partially on observations from optical sensors such as AVHRR and their accuracy is influenced by adverse weather conditions, including the presence of cloud cover. IMS derived lake ice products have been





widely used in monitoring global climate change (Duguay et al., 2013; Duguay et al., 2014; Duguay et al., 2015b); however, the IMS detected freeze onset was found to be too early for some lakes in northern Quebec, presumably due to misclassification by inclusion of coarse resolution satellite passive microwave observations during periods of prolonged cloud cover (Brown and Duguay, 2012; Brown et al., 2014); this may cause the low correlations between LIP and IMS CFO dates, as well as a delayed LIP CFO bias relative to IMS. In addition, the relatively coarse spatial resolution of AMSR observations limits capabilities for resolving lake ice conditions of finer scale water bodies. Space-borne and airborne optical-TIR and radar sensors are capable of improved delineation of smaller lakes and rivers (Chu et al., 2016), but at the expense of degraded temporal fidelity for regional and global applications.

As a proxy indicator of climate variability and change (Duguay et al., 2006), lake ice phenology variables and their changing trends are important for monitoring and understanding climate change and its feedbacks. As described above, only 5 of the 71 lakes examined showed statistically significant trends towards earlier WCI dates while no lake showed a significant later CFO trend. Earlier ice break-up events are signs of warmer spring conditions, which promote melting and breakup of lake ice and a lower surface albedo that absorbs more incoming solar radiation and further intensifies the rate of ice melt (Mishra et al., 2011). These results are also consistent with previous studies over Canada that found a general trend toward earlier springs and WCI dates particularly over western Canada but little change in isotherm and CFO dates in autumn (Duguay et al., 2006). Our results also indicate that lakes at higher latitudes are more likely to experience trends toward earlier spring ice break-up and shorter ICDe, which is consistent with enhanced warming trends at higher latitudes (Solomon et al., 2007; Deutsch et al., 2008). The above ice phenology trends coincide with regional SAT trends from ERA-Interim that show an average spring warming rate that is more than triple that of autumn, as well as stronger warming trends for higher latitudes.

Though the LIP lake ice phenology trends are generally consistent with regional climate warming (Magnuson et al., 2000; Solomon et al., 2007), further analysis based on a longer period of record is needed for distinguishing long-term climate trends from large inter-annual variability and periodic climate cycles, including the North Atlantic Oscillation (NAO), El-Niño Southern Oscillation (ENSO), and Pacific Decadal Oscillation (PDO) (Mishra et al., 2011).

## 5 Conclusions

Lake ice phenology is strongly influenced by variations in air temperature, while consistent long-term records of lake ice changes provide a sensitive climate change indicator (Magnuson et al., 2000; Weyhenmeyer et al., 2011). Continuous and accurate monitoring of lake ice dynamics is greatly needed for studies of global change and for monitoring lake ice impacts on ecosystems and infrastructure, especially for high-latitude and high-altitude regions. In this study, we developed a new automated algorithm for consistent daily retrieval of lake ice conditions over the Northern Hemisphere using similar 36.5 GHz H-polarized $T_b$ observations from AMSR-E and AMSR2 sensor records. The resulting 5 km resolution lake ice phenology record allows for daily monitoring of lake ice conditions without being significantly degraded by variations in



solar illumination or cloud and atmosphere contamination effects. In particular, the LIP record distinguished 71 large lakes that satisfied 20-day minimum ICDe and open water season algorithm thresholds; these lakes represent approximately 23.3% of the total surface area of large lakes (area $\geq$ 50 km$^2$) within the Northern Hemisphere domain. Smaller water bodies were excluded from the lake-wide analysis if the lakes had no pixels with complete (100%) open water coverage outside of a 5-km land buffer zone. The relatively coarse spatial resolution of AMSR observations limit capabilities for resolving finer scale water bodies, while the conservative lake selection criterion minimizes potential land contamination effects. The LIP derived lake ice conditions were found to be largely consistent with GLRIPD ground-based observations, with an average agreement of 95.4% for the four lakes examined. The LIP record also showed favourable correspondence with other lake CFO and WCI assessments defined from the CIS and IMS products for twelve large study lakes. The LIP, CIS and IMS differences were attributed to the different data sources and methods used to construct the different products, including differences in spatial and temporal resolutions of observations, and distinct nature of optical and microwave remote sensing. Though the design of the LIP algorithm, including the MTT method, helps to identify lake break-up/freeze-up events, while minimizing other $T_b$ disturbances from short-term weather events, atmosphere effects can still lead to retrieval errors, especially from persistent high atmosphere water vapour concentrations over high-latitude lakes in the summer. Based on the LIP record from 2002 to 2015, significant earlier melting of lake ice cover was detected for 5 of the 71 lakes examined in the Northern Hemisphere, while lakes at higher latitudes show a more evident warming trend toward earlier ice breakup and shorter ICDe than those at lower latitudes. As the operations from AMSR2 and similar sensors continue, the MTT algorithm will allow for automated retrieval and consistent monitoring of ice conditions for large Northern Hemisphere lakes into the future.

*Acknowledgements.* AMSR-E data are produced by Remote Sensing Systems and sponsored by the NASA Earth Science MEaSUREs DISCOVER Project and the AMSR-E Science Team. Data are available at www.remss.com. AMSR-E data and land cover classification maps were also provided courtesy of the National Snow and Ice Data Center (NSIDC). The AMSR2 L1R Tb data used for this study were provided courtesy of JAXA. The Global Lakes and Wetlands Database database is provided by the World Wildlife organization and created by the Center for Environmental Systems Research, University of Kassel, Germany. This work was conducted at the University of Montana with funding from NASA (NNX15AT74A).

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





**Table 1. Definitions of remotely sensed lake ice phenology variables from this study in relation to the other lake ice observational datasets used for the LIP validation assessment on a per pixel basis and for entire lakes.**

| Per pixel basis | | Entire lake | |
|---|---|---|---|
| **Terminology** | **Definition** | **Terminology** | **Definition** |
| Ice-on date | Day of year on which a pixel becomes totally ice-covered | Complete freeze over (CFO) date | Day of year when all pixels become totally ice-covered. |
| Ice-off date | Day of year on which a pixel becomes totally ice-free | Water clear of ice (WCI) date | Day of year when all pixels become totally ice-free |
| | | Ice cover duration (ICDe) | number of days between CFO and WCI |
| Associated dataset: LIP | | Associated datasets: LIP, IMS, CIS | |





**Table 2. Summary of the comparison results for water clear of ice (WCI) and complete freeze over (CFO) dates derived from AMSR Lake Ice Phenology (LIP), Canadian Ice Service (CIS) datasets for the period 2002-2015; and NOAA/IMS (IMS) for the period 2004-2015. $R_{A,B}$ denotes the correlation coefficient of datasets A and B; and $D_{A,B}$ is the average difference in unit day (A minus B) of WCI or CFO dates derived from datasets A and B.**

| Lake Name | Statistics of WCI date comparisons | | | | Statistics of CFO date comparisons | | | |
|---|---|---|---|---|---|---|---|---|
| | $R_{LIP,CIS}$ | $R_{LIP,IMS}$ | $D_{LIP,CIS}$ | $D_{LIP,IMS}$ | $R_{LIP,CIS}$ | $R_{LIP,IMS}$ | $D_{LIP,CIS}$ | $D_{LIP,IMS}$ |
| Great Bear | 0.90 | 0.94 | -2 | -3 | 0.90 | 0.54 | 0 | 3 |
| Great Slave | 0.85 | 0.91 | -6 | -5 | 0.72 | 0.63 | -2 | 3 |
| Smallwood | 0.66 | 0.62 | -6 | -4 | 0.70 | -0.08 | -4 | 1 |
| Nettiling | 0.91 | 0.84 | -9 | -4 | 0.92 | 0.33 | -2 | 10 |
| Dubawnt | 0.92 | 0.79 | -8 | 1 | 0.34 | 0.78 | -9 | -5 |
| Amadjuak | 0.89 | 0.80 | -7 | -4 | 0.87 | 0.27 | -2 | 6 |
| Wollaston | 0.95 | 0.87 | -5 | 0 | 0.66 | -0.42 | -7 | 9 |
| Baker | 0.80 | 0.70 | -10 | -1 | 0.60 | 0.26 | -7 | 0 |
| Kasba | 0.96 | 0.77 | -5 | 3 | 0.19 | 0.61 | -7 | 1 |
| Lesser Slave | 0.82 | 0.92 | -12 | -5 | 0.72 | 0.75 | -11 | -3 |
| Red Lake | 0.86 | 0.91 | -4 | 2 | 0.77 | 0.72 | -10 | -9 |
| Peter Pond | 0.85 | 0.95 | -7 | -1 | 0.88 | 0.54 | -5 | 0 |
| **Average** | **0.86** | **0.84** | **-7** | **-2** | **0.69** | **0.41** | **-6** | **1** |





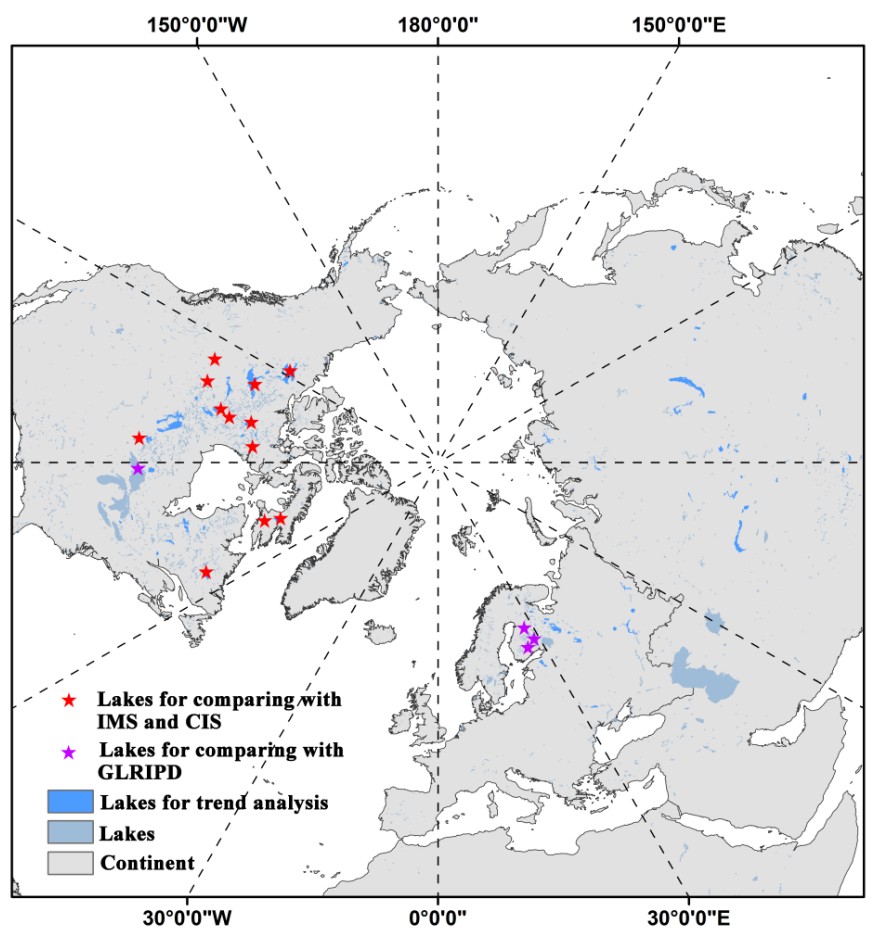

**Figure 1. Three sets of lakes selected for the LIP analysis in the Northern Hemisphere mid and high latitudes (≥30°N). The purple star symbols denote lakes used for evaluating LIP retrievals on a per pixel basis against GLRIPD ground-based observations, including Lake Superior in the USA; Lake Oulujarvi, Lake Haukivesi, and Lake Paijanne in Finland. The red star symbols denote**
10    **12 lakes (Great Bear Lake, Great Slave Lake, Smallwood Lake, Nettiling Lake, Dubawnt Lake, Amadjuak Lake, Wollaston Lake, Baker Lake, Kasba Lake, Lesser Slave Lake, Red Lake, Peter Pond Lake) used for the lake-wide comparisons between the LIP results from this study and other regional lake ice phenology records (IMS, CIS). The 71 lakes selected for assessing LIP trends over the 12-year satellite record are in bright blue.**





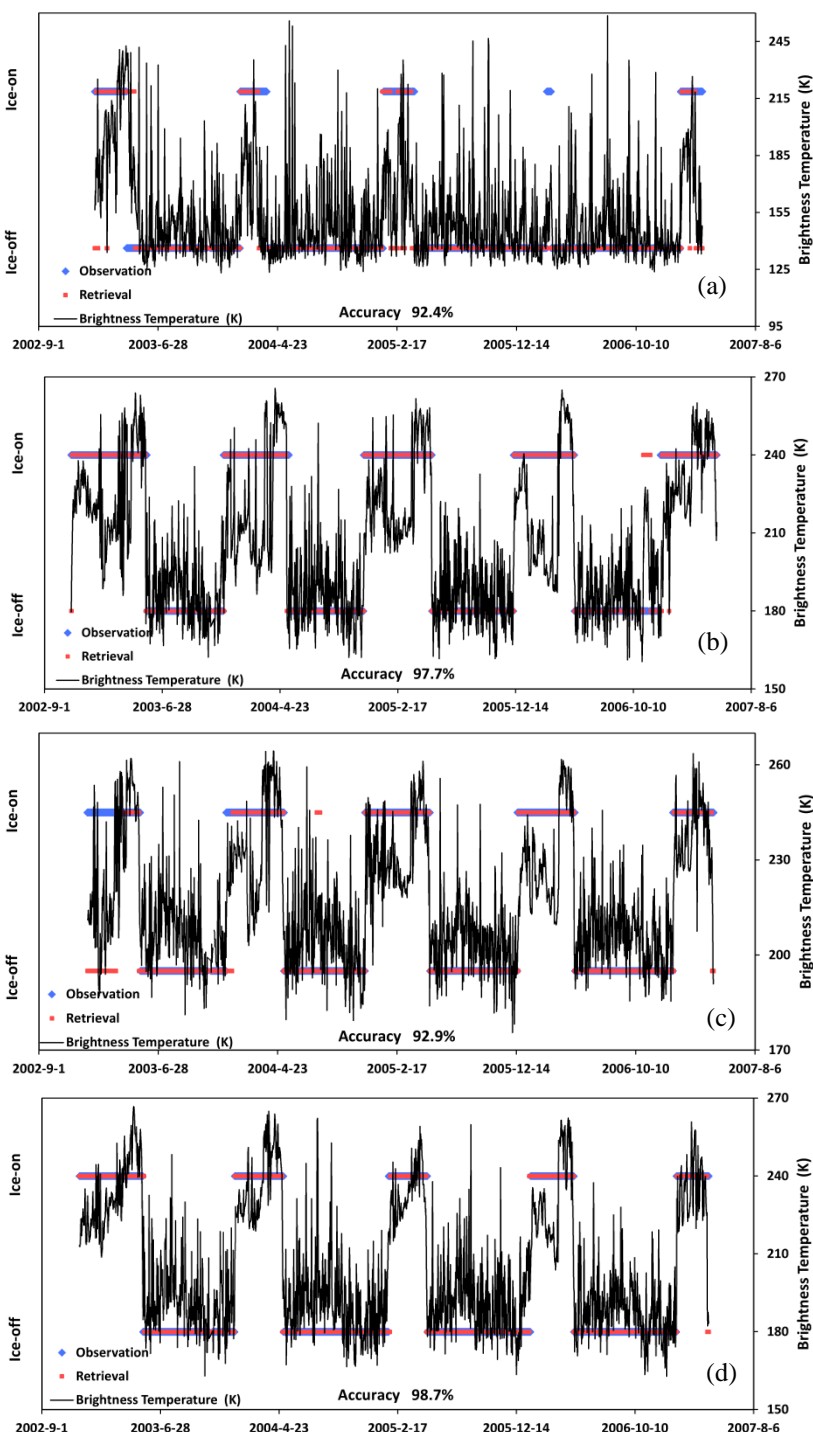

**Figure 2. Comparison of lake ice status for Lake Superior, USA (a), Lake Oulujarvi, Finland (b), Lake Haukivesi, Finland (c) and Lake Paijanne, Finland (d) derived from the Global Lake and River Ice Phenology Database (GLRIPD) (blue dots) and AMSR Lake Ice Phenology retrievals (LIP) (red dots). The AMSR 36.5 GHz H-Polarized daily $T_b$ retrievals used in the LIP algorithm are also plotted for reference (black line).**





**Figure 3. Comparisons between MODIS quick-look images (left column) and AMSR LIP results (right column) for Great Bear Lake (GBL) on Jun. 22 (a), Jun. 27 (b), Jul. 5 (c) and Jul. 8, 2013 (d).**




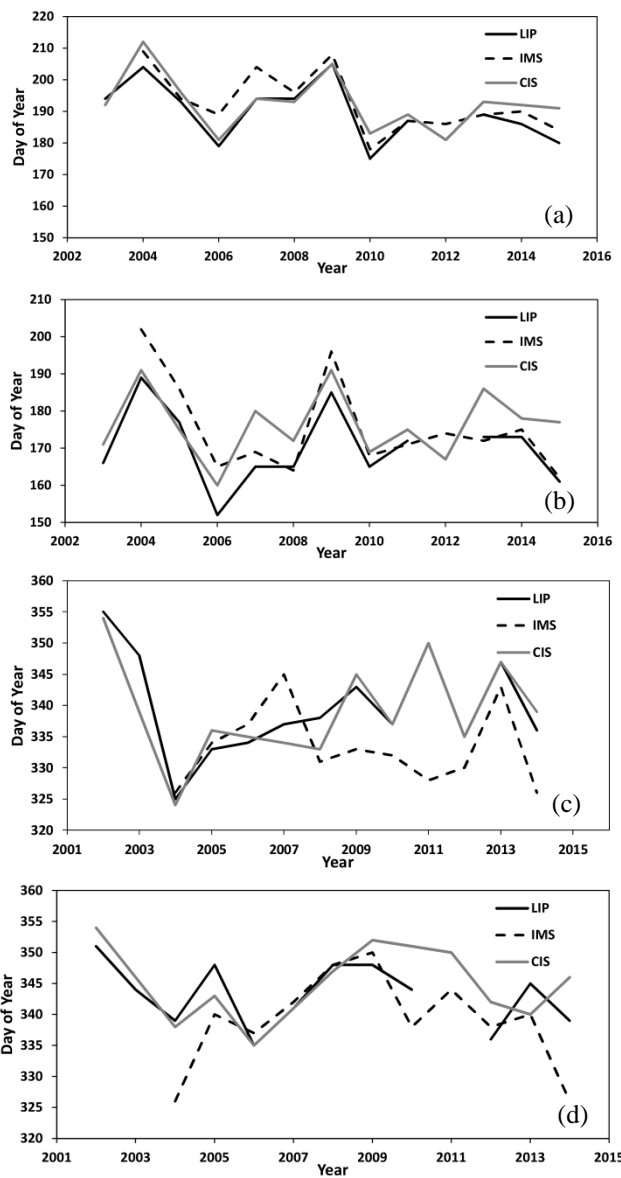

**Figure 4.** Comparisons of water clear of ice (WCI) dates for (a) Great Bear Lake and (b) Great Slave Lake, and complete freeze over (CFO) dates for (c) Great Bear Lake and (d) Great Slave Lake, derived from the AMSR Lake Ice Phenology (LIP) dataset developed in this study. The LIP results are compared against similar metrics derived from the NOAA/IMS (IMS) and Canadian Ice Service (CIS). Missing LIP data from 2011-2012 denotes the period between the end of AMSR-E operations and the start of the AMSR2 record.





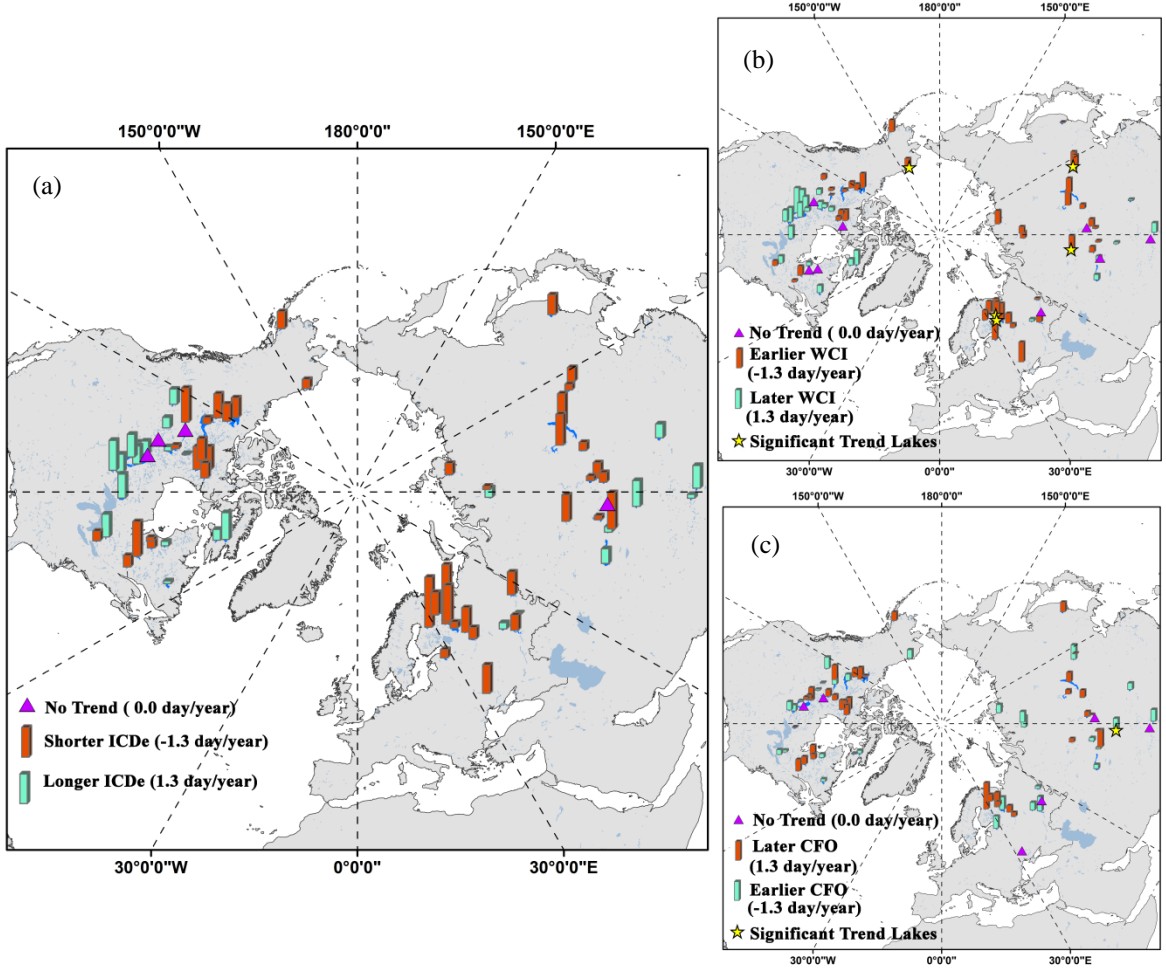

**Figure 5.** Changing trends of (a) ice cover duration (ICDe), (b) water clear of ice (WCI) dates and (c) complete freeze over (CFO)
5 dates of 71 lakes for the period 2002-2015. Lake changing trends are shown by bar symbols whose heights are proportional to the
trend magnitudes; the significant trend lakes are marked by yellow stars, while purple triangles denote lakes where no trend was
detected (rate of change is ~0.0 day/year).

