# Peer review of "Satellite Microwave Assessment of Northern Hemisphere Lake Ice Phenology from 2002 to 2015"

_The Cryosphere, 2016_

## Referee Comment (RC1) · Anonymous Referee #1 · 20 Oct 2016

The authors have developed an automated method for assessing timing of ice-on and ice-off for large lakes in the Northern Hemisphere using AMSR-E/2 passive microwave observations. Their results generally compare well with available observations from the ground and other satellite-derived records, such as from the NOAA 4-km IMS maps. Increasingly shorter ice duration for 43 of the 71 lakes was found, but results of only five lakes were statistically significant. Larger trends toward shorter ice duration were found at higher latitudes.

This is generally a very interesting paper and clearly written, though I have some comments. Please provide more detail regarding the implications of the 5-km spacing (in section 2.1.4), since the native resolutions of the AMSR-E/2 sensors are coarser than your product resolution.

It would be good to mention in the abstract that the automated method is consistent, whereas manual methods are not. This is mentioned later in the paper but is an important point with respect to climate-change studies.

Can you include information on the warming trends in the Northern Hemisphere during the 12-year study period, since warming is the likely driver of the shorter ice duration? In particular is there evidence of greater warming during the study period at the higher latitudes? If so, this would support the finding that the higher latitude lakes show a shorter ice duration (vs lower latitude lakes) during the 12-yr study period.

It is correctly stated that optical data (e.g., IMS, AVHRR and MODIS) can generally provide an update on ice conditions every few days (not daily due to clouds). How is your automated passive microwave algorithm superior? In other words, why is it important to have the update on ice conditions daily vs every few days? An added advantage of optical data is that the resolution is better than passive microwave. With the passive microwave you are only able to detect changes in the largest lakes.

I found the number of acronyms daunting to read and remember, though most if not all have been spelled out appropriately.

I found Table 2, and especially the caption, to be confusing.

Other comments: p.3, line 5 – I don't consider 1978 to present to be "long term" – please reconsider the wording p.3, line 23 – please consider using the word "including" versus "encompassing" p.3, line 30 – the paragraph starting on this line and ending on the next page, line 24 seems way too long; can you break it up into 2-3 paragraphs to increase the clarity? p.4, line 16 – consider using the word "represent" versus "encompass" p.5, section 2.1.3, and p.5, sentence on lines 25-27 - first paragraph; please check to see if you have already stated this p.5, line 25 – Lake Superior is in both the US and Canada; please fix p.6, line 14 – have you previously spelled out the acronym SAT? p.8, line 25 – last sentences of this paragraph – please provide a more thorough discussion of the uncertainties associated with gap filling over a period of 7 months

p.10, line 19 – consider using the word "show" versus "indicate" p.12, first paragraph – please break up into 2-3 paragraphs to increase clarity p.12, last sentence of first paragraph – how do you know that the LIP was incorrectly detected? p.13, paragraph 1 – please break up into two paragraphs to increase clarity Figure 1 – on the map, it can be hard to distinguish the two different shades of blue Figure 2 caption – Lake Superior is not only in the USA; please include Canada Figure 5 – the yellow stars do not show up well in figures 5b and 5c

---

## Referee Comment (RC2) · Anonymous Referee #2 · 23 Oct 2016

This manuscript presents a new methodology to apply the use of passive microwave imagery (AMSR-E/2) to create an automated daily ice phenology product utilized for monitoring freshwater lake ice. The daily temporal ability, unhindered by cloud and polar darkness allows for a global product at a resolution and accuracy that can of benefit climate modelling and global change studies. This manuscript is well written and a valuable contribution to the freshwater ice research field. The inclusion of cited literature is light in the introduction section and could use some strengthening; however overall, the manuscript is clear, concise and suitable for publication with only minor revision to the text and figures.

No particular scientific issues arise from this review. The validation of the new passive microwave based product is limited by the lack of existing observation data from lakes, however including the comparison to the CIS dataset and the IMS product strengthens

the validation sufficiently.

The following is a list of some minor typographical suggestions and some points to clarify the figures:

Line 11 : Previous studies have documented significantly earlier ice break-up between 1951 and 2000 for lakes in Canada (Duguay et al., 2006) and decreasing lake ice cover of Lake Ladoga in Europe during the last few decades (Karetnikov and Naumenko, 2008). – Consider adding more than 1 citation for each broad area of research

Line 16: various tendencies over specific lakes and . . . Such as?

Page 3 line 12: Recently, H-Polarized AMSR-E (Advanced Microwave Scanning Radiometer for EOS) Tb observations at 19 GHz were analyzed to determine ice phenology for GSL and Great Bear Lake (GBL), the two largest lakes in northern Canada (Kang et al., 2012). – again, consider using more than 1 citation for the work done here

Page 3 (line 30,31) - 4 (line 1) study domain – The explanation of the study purpose does not belong here.

Page 4, line 4 – arctic region "Global Warming" – consider a more scientific term. E.g. Arctic amplification effect? More extreme climate warming than lower latitudes?

Page 4, line 9 - Lake Superior is in Canada And USA.

Page 11 line 15 – ICDe? Was that defined previously? Why an e?

Figure 1 – needs more distinguishable blue colours for the trend analysis lakes. Lake Superior is listed as USA – Lake Superior is in Canada as well. Countries are listed for the GLRIPD lakes but not the other comparison lakes – why? Be consistent, either remove the countries for the GLRIPD lakes or add them for all lakes.

Figure 2 – clarify what the "dots" are – not clear initially what is going on this figure. Clearly indicated the "dots" represent ice cover vs. open water (?)

Figure 3 – This is a very unnatural projection for GBL, is there a specific reason to display this way? If so, it was unclear from the text. Consider reprojecting to a projection that is more realistic of the actual size/shape of the lake.

Figure 5 – ICDe? Again, why the e? b and c are too small to interpret the symbols / heights of the bars. Consider making all 3 the same size.

---

## Author Comment (AC1) · 1 Dec 2016

**Reply to RC1 for the manuscript "Satellite Microwave Assessment of Northern Hemisphere Lake Ice Phenology from 2002 to 2015" by Jinyang Du, John S. Kimball, Claude Duguay, Youngwook Kim, and Jennifer D. Watts**

Dear Anonymous Referee #1, thank you for your constructive comments on our manuscript. Please find below our responses to all of the review comments (in ***bold and italic***). The resulting manuscript changes are highlighted in blue.

***The authors have developed an automated method for assessing timing of ice-on and ice-off for large lakes in the Northern Hemisphere using AMSR-E/2 passive microwave observations. Their results generally compare well with available observations from the ground and other satellite-derived records, such as from the NOAA 4-km IMS maps. Increasingly shorter ice duration for 43 of the 71 lakes was found, but results of only five lakes were statistically significant. Larger trends toward shorter ice duration were found at higher latitudes.***

Reply: Thank you for the accurate summary.

***This is generally a very interesting paper and clearly written, though I have some comments. Please provide more detail regarding the implications of the 5-km spacing (in section 2.1.4), since the native resolutions of the AMSR-E/2 sensors are coarser than your product resolution.***

Reply:  The 5-km spacing is for facilitating analysis with alternative lake products derived with similar grid spacing, including the NOAA IMS 4-km daily snow and ice product, and a recently developed land surface fractional open water cover dataset, which was derived from AMSR-E/2 at 5 km gridding (Du et al., 2016).

We added in Section 2.1.4 the following sentences for improving clarity: "It is worth noting that the $T_b$ spatial gridding is posted at 5 km resolution while the original 36.5 GHz AMSR-E/2 observations have coarser native sensor footprints (~12 km for AMSR-E and 9 km for AMSR2). The finer grid spacing is intended to facilitate product comparisons and analyses with other alternative lake products derived at similar resolutions, including the NOAA IMS 4-km daily snow and ice product, and a land surface fractional open water cover dataset derived from AMSR-E/2 at 5 km resolution (Du et al., 2016). "

The following reference was added in the revision:

"Du, J., Kimball, J.S., Jones, L.A. and Watts, J.D.: Implementation of satellite based fractional water cover indices in the pan-Arctic region using AMSR-E and MODIS, Remote Sens. Environ., 184, 469-481, 2016."

***It would be good to mention in the abstract that the automated method is consistent, whereas manual methods are not. This is mentioned later in the paper but is an important point with respect to climate-change studies.***
Reply: Thank you for the suggestion. The first sentence of the abstract was revised accordingly as follows "A new automated method enabling consistent satellite assessment of seasonal lake ice phenology at 5-km resolution was developed for all lake pixels…".

*Can you include information on the warming trends in the Northern Hemisphere during the 12-year study period, since warming is the likely driver of the shorter ice duration? In particular is there evidence of greater warming during the study period at the higher latitudes? If so, this would support the finding that the higher latitude lakes show a shorter ice duration (vs lower latitude lakes) during the 12-yr study period.*

Reply: As analyzed in the manuscript using ERA-Interim reanalysis, the increase of surface air temperature (SAT) is more prominent at higher latitudes, which suggests greater warming during the study period for these regions. We further clarified the point in Section 3.3 of the revised manuscript as follows "Moreover, similar to the latitudinal pattern shown in the LIP-based analysis, the SAT increase in the spring is positively correlated with latitude (R = 0.33; p = 0.005) indicating greater warming during the study period at higher latitudes, while no SAT correlation with latitude is found for the autumn (R ~ 0.0)".

*It is correctly stated that optical data (e.g., IMS, AVHRR and MODIS) can generally provide an update on ice conditions every few days (not daily due to clouds). How is your automated passive microwave algorithm superior? In other words, why is it important to have the update on ice conditions daily vs every few days? An added advantage of optical data is that the resolution is better than passive microwave. With the passive microwave you are only able to detect changes in the largest lakes.*

Reply: We agree with the reviewer that optical remote sensing is of great importance for mapping lake ice especially at relatively fine resolutions. Complementary to optical remote sensing, routine and frequent observations from passive microwave remote sensing is also valuable and needed by climate studies and operational applications. This is because (a) microwave remote sensing may be the only available remote sensing technique for evaluating high-latitude lake ice processes in autumn and early winter when optical observations are obscured by darkness and poor weather conditions (Kang et al., 2012). For example, persistent and extensive cloud cover in autumn in Finland and Canada was found to prevent the use of optical satellite imagery to assess lake freeze-up timing (Maslanik et al., 1987). In operational snow and ice mapping systems such as IMS, microwave retrievals are the default observation when optical data is attenuated by clouds for several days (Helfrich et al., 2007); (b) lake ice information at enhanced temporal resolution provides improved ice initialization for numerical weather models and is expected to improve model prediction accuracies (Helfrich et al., 2007); (c) frequent data acquisition and complete time series of images are valuable to monitor ice variability and changes including transient lake ice disturbances (Jeffries et al.,2005); and supportive to applications such as hydroelectric power generation, navigation and winter transportation, and production and distribution of food and water (Schröter et al., 2005; Weyhenmeyer et al., 2011). Despite having coarser spatial resolution than optical sensors, the capability for consistent daily lake ice monitoring available from the passive microwave observations provides added precision for delineating long-term trends in ice on/off timing, which show more subtle changes than the larger characteristic year-to-year variability in ice cover. For clarifying the point, we revised the "Introduction" as follows:

"However, regional monitoring of lake ice dynamics from satellite optical-TIR sensors is constrained by signal degradation and data loss stemming from seasonal reductions in solar illumination at higher latitudes and persistent cloud cover, smoke and other atmospheric aerosol

contamination (Maslanik et al., 1987; Jeffries et al., 2005; Helfrich et al., 2007; Kang et al., 2012)."

"Alternatively, space-borne microwave radiometers have provided brightness temperature ($T_b$) observations since 1978 with relatively high temporal fidelity 5 (~1-2 days) especially at higher ($\geq$ 45˚N) latitudes. Frequent microwave radiometer data acquisition and complete time series of images are valuable to ice phenology studies and also supportive to improving numerical weather model predictions (Helfrich et al., 2007) and timely monitoring of lake ice events including transient ice disturbances (Jeffries et al., 2005). Despite having relatively coarser spatial resolution retrievals than optical-TIR sensors, the capability for consistent daily lake ice monitoring available from passive microwave observations provides added precision for delineating lake ice phenology trends, which may be much smaller than year-to-year ice cover variability."

We also added the following references in the revision:

"Jeffries, M. O., Morris, K., and Kozlenko, N.: Ice characteristics and processes, and remote sensing of frozen rivers and lakes in remote sensing in northern hydrology: Measuring environmental change, edited by: Duguay, C. R. and Pietroniro, A., Geophysical Monograph 163, American Geophysical Union, 63–90. 2005.

Maslanik, J. A. and Barry, R. G.: Lake ice formation and breakup as an indicator of climate change: Potential for monitoring using remote sensing techniques, The Influence of Climate Change and Climatic Variability on the Hydrologic Regime and Water Resources, International Association of Hydrological Sciences Press, IAHS Publ. No. 168, 153–161, 1987"

*I found the number of acronyms daunting to read and remember, though most if not all have been spelled out appropriately.*
Reply: To facilitate easier reading, we added an "Appendix" section in the revision and defined abbreviations and acronyms used in the manuscript as shown below. In addition, we used the abbreviation "AMSR-E/2" when discussing the combined AMSR-E and AMSR2 sensor records.

"

**Appendix**

List of Abbreviations and Acronyms

| | |
|---|---|
| AMSR2 | Advanced Microwave Scanning Radiometer 2 |
| AMSR-E | Advanced Microwave Scanning Radiometer for EOS |
| AMSR-E/2 | Advanced Microwave Scanning Radiometer for EOS and Advanced Microwave Scanning Radiometer 2 |
| AMSU | Advanced Microwave Sounding Unit |
| AVHRR | Advanced Very High Resolution Radiometer |
| CFO | complete freeze over |
| CIS | Canadian Ice Service |
| ERA-Interim | A global atmospheric model data reanalysis produced by the European Centre for |

                    Medium-Range Weather Forecasts
GBL              Great Bear Lake
GLRIPD       Global Lake and River Ice Phenology Database
GLWD           Global Lakes and Wetlands Database
GOES            Geostationary Operational Environmental Satellite
GSL              Great Slave Lake
IMS              Interactive Multisensor Snow and Ice Mapping System
LST              land surface temperature
MODIS         Moderate Resolution Imaging Spectroradiometer
MOD44W   MODIS 250 m land-water mask
MTT             Moving t-Test method
PFCs            PerFluorinated chemicals
*R*               correlation coefficient
RFI              Radio Frequency Interference
SAT              surface air temperature
SMMR         Scanning Multichannel Microwave Radiometer
SSM/I          Special Sensor Microwave Imager
$T_b$              brightness temperature
TIR             thermal infrared
VIIRS           Visible/Infrared Imaging Radiometer Suite
WCI            water clear of ice
”

*I found Table 2, and especially the caption, to be confusing.*
Reply: Table 2 caption was re-written to improve clarity as: "Table 2. Summary of the comparison results for water clear of ice (WCI) and complete freeze over (CFO) dates derived from AMSR Lake Ice Phenology (LIP) and Canadian Ice Service (CIS) datasets for the period 2002-2015, and the NOAA/IMS (IMS) dataset for the period 2004-2015. $R_{LIP,CIS/IMS}$ denotes the correlation coefficient between the LIP and CIS/IMS datasets; $D_{LIP,CIS/IMS}$ is the average difference (unit: day) in WCI or CFO dates calculated by LIP minus CIS/IMS. "

*Other comments:*
*p.3, line 5 – I don't consider 1978 to present to be "long term" –please reconsider the wording*
Reply: We deleted "long term" and the sentence was revised as "Alternatively, space-borne microwave radiometers have provided brightness temperature ($T_b$) observations since 1978 with relatively high temporal fidelity 5 (~1-2 days) especially at higher ($\geq 45˚N$) latitudes"

*p.3, line 23 – please consider using the word "including" versus "encompassing"*
Reply: The sentence was revised as "…observations including both AMSR-E (June 2002 to September 2011) and AMSR2 (June 2012 to December 2015) satellite sensor records."

*p.3, line 30 – the paragraph starting on this line and ending on the next page line 24 seems way too long; can you break it up into 2-3 paragraphs to increase the clarity?*
Reply: As suggested by the reviewer, the paragraph was divided into two parts with the second paragraph starting from the sentence "Finally, regional LIP trends were assessed over the 12-year (2002-2015) satellite record…".

***p.4, line 16 – consider using the word "represent" versus "encompass"***
Reply: The sentence was revised as "The lakes selected represent …"

***p.5, section 2.1.3, and p.5, sentence on lines 25-27 - first paragraph; please check to see if you have already stated this***
Reply: The lake names "Lake Superior, Lake Oulujarvi, Lake Haukivesi and Lake Paijanne" appeared in Section 2.1.3 were previously stated in the first paragraph of page 4. In the revision, this repeat in Section 2.1.3 was deleted as shown below:
"Only four lakes were selected for the LIP comparisons due to a predominance of ice observations from smaller lakes in the GLRIPD database"

***p.5, line 25 – Lake Superior is in both the US and Canada; please fix***
Reply: We made corrections throughout the manuscript. The revisions include "Lake Superior in the USA and Canada" in paragraph 1 of page 4; and the captions of Fig.1 ("Lake Superior in the USA and Canada") and Fig.2 ("Lake Superior, USA and Canada").

***p.6, line 14 – have you previously spelled out the acronym SAT?***
Reply: Yes, please refer to the first sentence of the same paragraph as also listed below "In addition, ERA-Interim (Dee et al., 2011) quarter-degree reanalysis surface air temperature (SAT) data …". SAT is also defined in the Appendix section of the revised manuscript.

***p.8, line 25 – last sentences of this paragraph – please provide a more thorough discussion of the uncertainties associated with gap filling over a period of 7 months***
Reply: As stated in the first paragraph of page 3 "Alternatively, space-borne microwave radiometers have provided brightness temperature ($T_b$) observations since 1978 with relatively high temporal fidelity 5 (~1-2 days) especially at higher ($\geq$ 45˚N) latitudes", AMSR-E/2 daily temporal coverage is generally obtainable for high-latitudes but not for middle and low latitudes. Therefore, the missing daily $T_b$ retrievals especially for the middle and low latitudes were "gap-filled through temporal linear interpolation" (line 26, page 8). For the 7-month period without AMSR-E and AMSR2 observations, no retrievals were carried out as stated on lines 23-25 of page 8, "The above lake ice detection process was carried out for each $T_b$ time series from AMSR-E and AMSR2 separately because of the 7-month gap (Oct 4, 2011 – May 18, 2012) in the observation records between the two sensors". To avoid confusion, the sentences (line 23-28, page 8) were re-arranged and revised as follows:

"For running the algorithm, missing daily $T_b$ retrievals were obtained through temporal linear interpolation of adjacent successful $T_b$ retrievals acquired from the same ascending orbits. However, only the lake ice detection results corresponding to the actual satellite observations were output for further analysis. The above lake ice detection process was carried out for each $T_b$ time series from AMSR-E and AMSR2 separately because of the 7-month gap (Oct 4, 2011 – May 18, 2012) in the observation records between the two sensors."

***p.10, line 19 – consider using the word "show" versus "indicate"***
Reply: The sentence was revised as "…both MODIS and LIP show remaining ice cover on the western edge of…"

***p.12, first paragraph– please break up into 2-3 paragraphs to increase clarity***
Reply: As suggested by the reviewer, the paragraph was divided into two parts with the second paragraph starting from "Differences between the LIP and GLRIPD results can be attributed to several factors."

***p.12, last sentence of first paragraph – how do you know that the LIP was incorrectly detected?***
Reply: According to Finnish Meteorological Institute (http://en.ilmatieteenlaitos.fi/seasons-in-finland), summer in Finland usually begins in late May and lasts until mid-September with the mean daily temperature consistently above 10 ℃. It is unlikely that ice cover was present on July 30, 2004, which was in the mid-summer. To be more accurate, the sentence was re-written as "For example, the LIP detected ice-on conditions for Lake Haukivesi Finland in mid-summer (July 30, 2004) (Fig.2 c), which is likely incorrect and may be due to increased atmosphere water vapor concentrations under warm summer conditions, resulting in a large $T_b$ increase similar to a seasonal freeze-up event."

***p.13, paragraph 1 – please break up into two paragraphs to increase clarity***
Reply: As suggested by the reviewer, the paragraph was divided into two parts with the second paragraph starting from "In addition, the relatively coarse spatial resolution of …".

***Figure 1 – on the map, it can be hard to distinguish the two different shades of blue***
Reply: We re-plotted Fig.1 with more distinguishable blue colors in the revised manuscript as also shown below:

[Figure]

***Figure 2 caption – Lake Superior is not only in the USA; please include Canada***
Reply: The correction was made throughout the manuscript. The revisions include paragraph 1 of page 4 ("Lake Superior in the USA and Canada") and the captions of Fig.1 ("Lake Superior in the USA and Canada") and Fig.2 ("Lake Superior, USA and Canada").

***Figure 5 – the yellow stars do not show up well in figures 5b and 5c***
Reply: Reviewer 2 has similar concerns. As suggested by reviewer 2, we made Fig.5 (a), (b) and (c) the same size to improve clarity as also shown below.

[Figure]

[Figure]

---

## Author Comment (AC2) · 1 Dec 2016

**Reply to RC2 for the manuscript "Satellite Microwave Assessment of Northern Hemisphere Lake Ice Phenology from 2002 to 2015" by Jinyang Du, John S. Kimball, Claude Duguay, Youngwook Kim, and Jennifer D. Watts**

Dear Anonymous Referee #2, thank you for your helpful comments on our manuscript. Please find below our responses to all the comments (in ***bold and italic***). The changes on the manuscript were highlighted in blue.

***This manuscript presents a new methodology to apply the use of passive microwave imagery (AMSR-E/2) to create an automated daily ice phenology product utilized for monitoring freshwater lake ice. The daily temporal ability, unhindered by cloud and polar darkness allows for a global product at a resolution and accuracy that can of benefit climate modelling and global change studies. This manuscript is well written and a valuable contribution to the freshwater ice research field. The inclusion of cited literature is light in the introduction section and could use some strengthening; however overall, the manuscript is clear, concise and suitable for publication with only minor revision to the text and figures.***

***No particular scientific issues arise from this review. The validation of the new passive microwave based product is limited by the lack of existing observation data from lakes, however including the comparison to the CIS dataset and the IMS product strengthens the validation sufficiently.***
Reply: Thank you for the comments.

***The following is a list of some minor typographical suggestions and some points to clarify the figures:***

***Line 11 : Previous studies have documented significantly earlier ice break-up between 1951 and 2000 for lakes in Canada (Duguay et al., 2006) and decreasing lake ice cover of Lake Ladoga in Europe during the last few decades (Karetnikov and Naumenko, 2008). – Consider adding more than 1 citation for each broad area of research***
Reply: As suggested by the reviewer, the sentences were revised with more citations added as follows: "Previous studies have documented significantly earlier ice break-up between 1950s and 2000s for lakes in Canada (Duguay et al., 2006; Latifovic and Pouliot, 2007; Prowse et al., 2011) and decreasing ice cover duration of Eurasia lakes during the last few decades (Vuglinsky and Gronskaya, 2006; Karetnikov and Naumenko, 2008; Prowse et al., 2011)."

The added references were listed below:
"Vuglinsky, V.S. and Gronskaya, T.P.: Changing of rivers and lakes ice regime within the Russian territory and their possible consequences for economy, Modern problems of hydrometeorology, 229-245, 2006.

Prowse, T., Alfredsen, K., Beltaos, S., Bonsal, B., Duguay, C., Korhola, A., McNamara, J., Pienitz, R., Vincent, W.F., Vuglinsky, V. and Weyhenmeyer, G.A.: Past and future changes in Arctic lake and river ice, Ambio, 40(1), 53-62, 2011."

*Line 16: various tendencies over specific lakes and : : : Such as?*
Reply: The sentences were revised to improve clarity as follows:
"Despite a general tendency for later freezing and earlier break-up in the Northern Hemisphere (Magnuson et al., 2000), various tendencies including earlier ice formation and later ice break-up over specific lakes and time periods may exist. For example, observations from satellite altimetry and radiometry over 1992–2004 for Lake Baikal showed a tendency for colder winters, with earlier ice formation, later ice break-up, and ice duration increase (Kouraev et al., 2007a, 2007b)."

*Page 3 line 12: Recently, H-Polarized AMSR-E (Advanced Microwave Scanning Radiometer for EOS) Tb observations at 19 GHz were analyzed to determine ice phenology for GSL and Great Bear Lake (GBL), the two largest lakes in northern Canada (Kang et al., 2012). – again, consider using more than 1 citation for the work done here*
Reply: The AMSR-E lake ice detection method described here was mainly developed in (Kang et al., 2012) and extended in (Kang et al., 2014). As suggested by the reviewer, additional citation was listed as below: "Recently, H-Polarized AMSR-E (Advanced Microwave Scanning Radiometer for EOS) $T_b$ observations at 19 GHz were analyzed to determine ice phenology for GSL and Great Bear Lake (GBL), the two largest lakes in northern Canada (Kang et al., 2012; Kang et al., 2014)."

*Page 3 (line 30, 31) - 4 (line 1) study domain – The explanation of the study purpose does not belong here.*
Reply: We accepted the reviewer's comment and deleted the sentence "Accurate monitoring of seasonal land freeze/thaw and lake freeze-up/break-up events which are widespread in the Northern Hemisphere has been recognized as an essential component for understanding interactions and feedbacks between regional ecosystems and climate change (Duguay et al., 2006; Kim et al., 2012; Du et al., 2015a)".

*Page 4, line 4 – arctic region "Global Warming" – consider a more scientific term. E.g. Arctic amplification effect? More extreme climate warming than lower latitudes?*
Reply: As suggested by the reviewer, the sentence was revised as follows "The domain (Fig.1) includes the high northern pan-Arctic region and high altitude Qinghai-Tibetan Plateau, which are data-sparse but strongly sensitive to the Arctic amplification effect (Serreze and Francis, 2006; Woo et al., 2007) and/or elevation-dependent warming (Wang et al., 2011; Mountain Research Initiative EDW Working Group, 2015)."

The following references were added in the revision:
"Serreze, M. C., and Francis, J. A.: The Arctic amplification debate, Clim. Chang., 76(3-4), 241-264, 2006.

Mountain Research Initiative EDW Working Group: Elevation-dependent warming in mountain regions of the world, Nat. Clim. Chang., 5(5): 424-430, 2015."

*Page 4, line 9 - Lake Superior is in Canada And USA.*
Reply: We made corrections throughout the manuscript. The revisions include paragraph 1 of page 4 ("Lake Superior in the USA and Canada") and the captions of Fig.1 ("Lake Superior in the USA and Canada") and Fig.2 ("Lake Superior, USA and Canada").

***Page 11 line 15 – ICDe? Was that defined previously? Why an e?***
Reply: The ICDe abbreviation was defined in Table 1 and was adopted from previous studies (Kang et al., 2012; Duguay et al., 2015a). "e" here represents "entire lake". To be clearer, ICDe in Table 1 was re-defined as "Ice cover duration of entire lake (ICDe)".

***Figure 1 – needs more distinguishable blue colours for the trend analysis lakes.***
Reply: As suggested by the reviewer, Fig.1 was re-plotted with more distinguishable blue colors as shown below.

[Figure]

***Lake Superior is listed as USA – Lake Superior is in Canada as well.***
Reply: We made corrections throughout the manuscript. The revisions include paragraph 1 of page 4 ("Lake Superior in the USA and Canada") and the captions of Fig.1 ("Lake Superior in the USA and Canada") and Fig.2 ("Lake Superior, USA and Canada").

*Countries are listed for the GLRIPD lakes but not the other comparison lakes – why? Be consistent, either remove the countries for the GLRIPD lakes or add them for all lakes.*
Reply: As suggested by the reviewer, we added the country names for the other comparison lakes in the caption of Fig.1 and section 2.1.1 as follows:

"… The red star symbols denote 12 lakes (Great Bear Lake, Great Slave Lake, Smallwood Lake, Nettiling Lake, Dubawnt Lake, Amadjuak Lake, Wollaston Lake, Baker Lake, Kasba Lake, Lesser Slave Lake, and Peter Pond Lake in Canada; Red Lake in USA) …"

"And 12 North American lakes (GBL, GSL, Smallwood Lake, Nettiling Lake, Dubawnt Lake, Amadjuak Lake, Wollaston Lake, Baker Lake, Kasba Lake, Lesser Slave Lake, and Peter Pond Lake in Canada; Red Lake in USA)…"

*Figure 2 – clarify what the "dots" are – not clear initially what is going on this figure. Clearly indicated the "dots" represent ice cover vs. open water (?)*
Reply: The blue/red dots represent GLRIPD/LIP derived ice conditions as indicated by their Y-axis positions for the dates described by their X-axis coordinates. To be clearer, caption of Figure 2 was revised as follows:

"Figure 2. Comparison of lake ice status for Lake Superior, USA (a), Lake Oulujarvi, Finland (b), Lake Haukivesi, Finland (c) and Lake Paijanne, Finland (d) derived from the Global Lake and River Ice Phenology Database (GLRIPD) and AMSR-E/2 Lake Ice Phenology retrievals (LIP). The AMSR 36.5 GHz H-Polarized daily Tb retrievals used in the LIP algorithm are also plotted for reference (black line). The blue/red dots represent GLRIPD/LIP derived ice conditions as indicated by their Y-axis positions for the dates described by their X-axis coordinates."

*Figure 3 – This is a very unnatural projection for GBL, is there a specific reason to display this way? If so, it was unclear from the text. Consider reprojecting to a projection that is more realistic of the actual size/shape of the lake.*
Reply: Fig. 3 was plotted under EASE-GRID version 2 to be consistent with the AMSR-E/2 gridded $T_b$ data as described in Section 2.1.4. The EASE-GRID projection is a standard format for cryosphere products including sea ice retrievals derived from passive microwave sensors AMSR-E and SSM/I (https://nsidc.org/data/seaice/pm.html). To be clearer, we revised the caption of Fig. 3 as follows:

"Figure 3. Comparisons between MODIS quick-look images (left column) and AMSR LIP results (right column) for Great Bear 5 Lake (GBL) on Jun. 22 (a), Jun. 27 (b), Jul. 5 (c) and Jul. 8, 2013 (d). The images are in the EASE-GRID version 2 polar projection format, consistent with the underlying AMSR-E/2 gridded $T_b$ dataset used for the LIP classification."

*Figure 5 – ICDe? Again, why the e?*
Reply: The ICDe was defined in Table 1 and previous studies (Kang et al., 2012; Duguay et al., 2015a). "e" here represents "entire lake". As replied previously, ICDe in Table 1 was re-defined as "Ice cover duration of entire lake (ICDe)".

*b and c are too small to interpret the symbols / heights of the bars. Consider making all 3 the same size.*

Reply: We followed the reviewer's suggestion and made Fig.5 (a), (b) and (c) the same size to improve its clarity as shown below.

[Figure]

[Figure]